# DPFormer: Learning Differentially Private Transformer on Long-Tailed Data

## Abstract

The Transformer has emerged as a versatile and effective architecture with broad applications. However, it still remains an open problem how to efficiently train a Transformer model of high utility with differential privacy guarantees. In this paper, we identify two key challenges in learning differentially private Transformers, i.e., heavy computation overhead due to per-sample gradient clipping and unintentional attention distraction within the attention mechanism. In response, we propose DP-Former, equipped with Phantom Clipping and Re-Attention Mechanism, to address these challenges. Our theoretical analysis shows that DPFormer can reduce computational costs during gradient clipping and effectively mitigate attention distraction (which could obstruct the training process and lead to a significant performance drop, especially in the presence of long-tailed data). Such analysis is further corroborated by empirical results on two real-world recommendation datasets with varying degrees of long-tailedness, showing its significant improvement in terms of efficiency and effectiveness.

## 1 Introduction

Differentially private deep learning has made remarkable strides, particularly in domains such as image classification (Tramer & Boneh, 2021; Golatkar et al., 2022; De et al., 2022) and natural language processing (Yu et al., 2022; Li et al., 2022b; He et al., 2023). This success can be largely attributed to the availability of extensive pre-trained models, offering robust and diverse foundations for further learning. However, such reliance on vast, pre-existing datasets poses a significant challenge when these resources are not accessible or relevant. This hurdle becomes particularly pronounced when it is necessary to train differentially private Transformers using only domain-specific data gathered from real-world scenarios rather than generic, large-scale datasets.

One relevant real-world scenario is privacy-preserving commercial recommender systems, where a transformer model needs to be trained differentially privately on users' historical behaviors for sequential prediction and large-scale public datasets or pre-existing pre-trained models do not exist.

The challenges posed by this scenario can be summarized into two key hurdles. The first one stems from the inherent nature of real-world data, which typically follows a long-tailed distribution, where a small fraction of data occur frequently, while a majority of data appear infrequently. This poses the intrinsic hardness of high-utility differentially private training, which, based on its sample complexity (Dwork et al., 2009), necessitates a *sufficiently* large volume of data to discern general patterns without resorting to the memorization of individual data points (Carlini et al., 2019; Feldman, 2020). Our theoretical analysis further shows that during differentially private training of Transformers, attention scores tend to be skewed by long-tailed tokens (i.e., tokens with fewer occurrences), therefore leading to huge performance drops. The second hurdle arises from the resource-intensiveness of deep learning with differential privacy, which is primarily due to the requirement of clipping per-sample gradient. This requirement not only complicates the training process but also places a significant computational burden, especially when resources are limited, which is typical for mobile devices.

To address these issues, we propose DPFormer (Figure 1), a methodology for enabling efficient and effective training of differentially private Transformers. Specifically, DPFormer consists of two key parts, i.e., Phantom Clipping and Re-Attention Mechanism. Phantom Clipping inherit the basic idea of Ghost Clipping (Li et al., 2022b), that is, obtaining the per-sample gradient norm without the need for instantiating the per-sample gradient. Our Phantom Clipping generalize this technique to the

shared embedding layer (among the input layer and output layer, which is the standard practice of training transformer or other embedding-based models) and provides speedup *for free* by leveraging the sparsity nature of the input. Re-Attention Mechanism aims to improve the effectiveness of private training in the presence of long-tailed data by mitigating the attention distraction phenomenon due to private training on long-tailed data. Experiments on public real-world recommendation datasets with varying degrees of long-tailedness show the significant improvement achieved by DPFormer in terms of efficiency and effectiveness.

## 2 PRELIMINARIES

**Problem Setting: Sequential Prediction.** Since Transformers are designed to predict the next token in an autoregressive manner, we focus our evaluation on sequential prediction tasks, where each training sample consists of a sequence of tokens[1] Given the preceding tokens $[s_1, s_2, ..., s_{t-1}]$, the task is to predict the next token $s_t$. Note that in practice (as is also the case for all our datasets), training data is typically long-tailed, in the sense that a small number of tokens occur quite frequently while others have fewer occurrences. Our goal is to train a Transformer with DP-SGD (Abadi et al., 2016) such that it can predict the next token accurately while preserving differential privacy.

**Definition 2.1.** $(\varepsilon, \delta)$**-Differential Privacy (DP) (Dwork et al., 2006; 2014):** A randomized mechanism $\mathcal{M} : \mathcal{D} \to \mathcal{R}$ satisfies $(\varepsilon, \delta)$-differential privacy if for any two datasets $\mathcal{D}, \mathcal{D}' \in$ Domain$(\mathcal{M})$ that differ in one record and for all $S \in$ Range$(\mathcal{M})$ it holds that $\Pr(\mathcal{M}(\mathcal{D}) \in \mathcal{S}) \leq e^{\varepsilon} \Pr(\mathcal{M}(\mathcal{D}') \in \mathcal{S}) + \delta$.

One desirable property of DP is that it ensures privacy (in terms of $\varepsilon$ and $\delta$) under composition. Based on this property, DP-SGD (Abadi et al., 2016) injects calibrated Gaussian noise into model gradients in each training step to achieve differential privacy as follows,

$$\boldsymbol{G} = \frac{1}{B} \left( \sum_{i=1}^{B} \boldsymbol{g}_i \cdot \text{Clip}_\text{C}(\|\boldsymbol{g}_i\| + \sigma_\text{dp} \cdot \mathcal{N}(0, \mathbf{I})) \right), \tag{1}$$

where $G$ is the averaged gradient among the minibatch, $\boldsymbol{g}_i$ is the gradient of the $i$-th sample in the minibatch of size $B$, $C$ is the clipping norm, $\text{Clip}_\text{C}(\|g_i\|) = \min(C/\|g_i\|, 1)$, ensuring that the sensitivity of the averaged gradient $G$ is bounded by $\Delta_G \leq \|g_i \cdot \text{Clip}(\|g_i\|)\| \leq C$. dp is the noise multiplier derived from privacy accounting tools (Balle et al., 2018; Wang et al., 2019).

The detailed discussion of related work is in Appendix A.1.

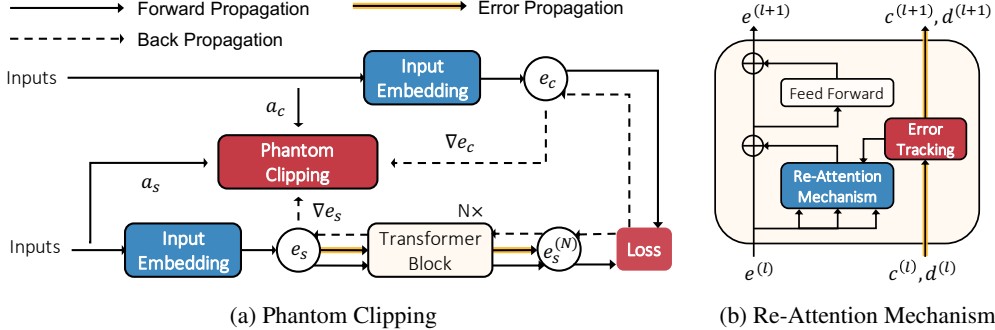

(a) Phantom Clipping          (b) Re-Attention Mechanism

Figure 1: The DPFormer - methodology overview. **Left:** Phantom Clipping allows parameter sharing of the embedding layer while providing additional speedup. **Right:** Re-Attention Mechanism aims to mitigate the attention distraction during private training and thereby improves the model performance.

---

[1]In this paper, we will use 'token' to denote the discrete unit within the input sequence and 'vocabulary size' to represent the total count of relevant entities, generalizing their definitions associated with language modeling.

## 3 PHANTOM CLIPPING

### 3.1 MOTIVATION: PARAMETER SHARING AS A FORM OF INDUCTIVE BIAS

The method of clipping per-sample gradient (Goodfellow, 2015) without instantiating per-sample gradient has shown considerable advantage (Li et al., 2022b) in training Transformer models using DP-SGD as compared to other libraries or implementations (for instance, Opacus (Yousefpour et al., 2021), JAX (Subramani et al., 2021)). The current limitation is that such idea does not support parameter sharing (i.e., the practice that ties the parameter of the input embedding and the output embedding layer together).

To show the importance of parameter sharing when training with DP-SGD, we conduct experiments under the following three settings: (1) parameter sharing of the embedding layer, which aligns with the standard treatment in Transformer; (2) no parameter sharing; and (3) no parameter sharing coupled with a reduced embedding dimension by half. Note that the third setting is included to account for the potential impact of model dimension on accuracy in private training, given the difference in the number of parameters between models with and without parameter sharing. Model performance across different hyperparameters is shown in Figure. 2. The consistency and significance of the performance improvement brought by parameter sharing during private training are not hard to perceive. The essence of embedding sharing lies in the assumption that, by tying the embedding of the input and output layers, the representation of each token remains consistent throughout its retrieval. This inductive bias enhances the statistical efficiency of the model, enabling improved generalization. When training with DP-SGD on limited training data, the model must independently uncover this relationship from the noisy gradients with a low signal-to-noise ratio, heightening the convergence challenge.

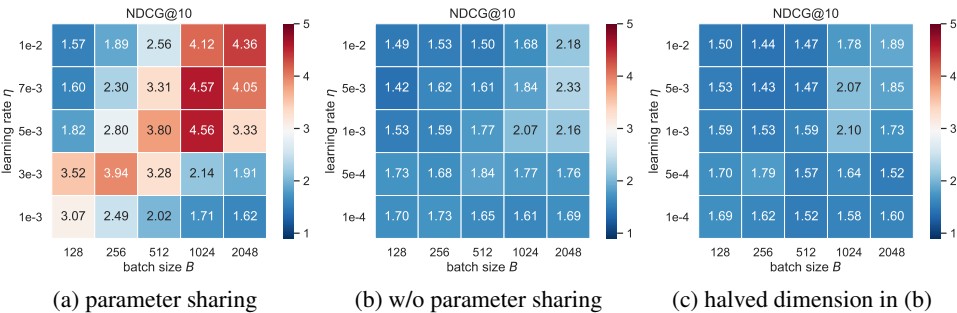

(a) parameter sharing (b) w/o parameter sharing (c) halved dimension in (b)

Figure 2: Numbers are NDCG(%)@10 (higher is better) of the privately trained model (with $\varepsilon$ set to 5) on MovieLens (Figure 5). Parameter sharing for the embedding layer yields consistent and significant performance gains over the non-sharing setting in private training. The optimal hyperparameter configuration is always using a large batch size (with a large learning rate).

### 3.2 EFFICIENT PER-SAMPLE GRADIENT NORM COMPUTATION WITH PARAMETER SHARING

In this section, we present *Phantom Clipping*, a technique for efficient private training of Transformers without the need for instantiating per-sample gradient. Besides additionally supporting parameter sharing compared with Ghost Clipping (Li et al., 2022b), Phantom Clipping provides further speedup (for free) by leveraging the sparsity of the input, which achieves significant gains when the efficiency issue associated with the embedding layer is the bottleneck.

Recall that the computational bottleneck of gradient clipping in Equation (1) lies in the calculation of the per-sample gradient norm i.e., $\|g_i\|$. As the L2 norm of a vector can be decomposed cross arbitrary dimensions, for example, $\|(a, b, c)\| = \|(\|a\|, \|(b, c)\|)\|$. It suffices to consider the per-sample gradient norm $\|g_{i,E}\|$ of the embedding layer $E$ because the disparity due to parameter sharing lies solely in the shared embedding, and other layers can be handled akin to Ghost Clipping. After obtaining the gradient norm via $\|g_i\| = \|(\|g_{i,E}\|, \|g_{i,-E}\|)\|$, the next step is to scale the gradient by a factor of $\text{Clip}_C(\|g_i\|)$ to bound its sensitivity. This can either be accomplished by re-scaling the loss $\mathcal{L}_i$ through this factor, followed by a second backpropagation (Lee & Kifer, 2021), or by manually scaling the gradient as demonstrated by (Bu et al., 2022b).

Therefore, the challenge of evaluating $\|g_{i,E}\|$ efficiently without instantiating $g_{i,E}$ stems from the non-plain feed-forward (and symmetrically, backward propagation) topology caused by parameter sharing. See Figure 1a for a visual illustration, where the shared embedding leads to two branches of the backpropagation.

**Claim 3.1.** *(**Phantom Clipping**) Let $L$ and $M$ be the input length and vocabulary size, respectively. Let $a_{i,s} \in \{0,1\}^{L \times M}$ (or $a_{i,c} \in \{0,1\}^{M \times M}$) be the one-hot encodings of the input sequence $s_i$ (or those of the candidate tokens for the output probability) in a minibatch. Let $e_{i,s} \in \mathbb{R}^{L \times d}$ (or $e_{i,c} \in \mathbb{R}^{M \times d}$) be output of the (shared) embedding layer $E$ when fed into $a_{i,s}$ (or $a_{i,c}$). Then the norm of the per-sample gradient with respect to $E$ can be efficiently evaluated as*

$$\|g_{i,E}\| = \left( \langle a_{i,s} a_{i,s}^T, \nabla e_{i,s} \nabla e_{i,s}^T \rangle^2 + \|\nabla e_{i,c}\|^2 + 2 \cdot \langle \nabla e_{i,s}, a_{i,s}^T \nabla e_{i,c} \rangle \right)^{\frac{1}{2}}, \qquad (2)$$

*where $\nabla e_{i,s} := \partial \mathcal{L}_i / \partial e_{i,s} \in \mathbb{R}^{L \times d}$, $\nabla e_{i,c} := \partial \mathcal{L}_i / \partial e_{i,c} \in \mathbb{R}^{M \times d}$, and $\langle \cdot, \cdot \rangle$ is the inner product of two matrices being of the same shape.*

The full version and the derivation is deferred to Appendix A.2. Note that besides supporting parameter sharing, our phantom clipping actually provides additional speedup by leveraging the sparsity of input via Equation (16). We first study the additional memory footprint required by the embedding layer. Due to the storage of $a_s a_s^T$ (and $\nabla e_s \nabla e_s^T$) $\in \mathbb{R}^{B \times L \times L}$ in the first term of Equation (16) (note that $a^T \nabla e_c$ is merely an indexing operation, requiring no additional memory), *Phantom Clipping* has overhead memory complexity of $O(BL^2)$. As a comparison, Ghost Clipping has a memory complexity of $O(BT^2)$ when the input to the layer $a_i$ has the shape of $\mathbb{R}^{T \times \cdot}$. Hence its memory complexity for the two embedding layers is $O(BM^2 + BL^2)$ where $M$ is the vocabulary size. The relative speedup is therefore $1 + O((M/L)^2)$ (recall that $(M/L)^2 \gg 1$ due to the sparsity of input, i.e., the input length is greatly smaller than the vocabulary size).

**Overall Speedup.** The total memory overhead comprises multiple additive components, each corresponding to a specific layer, i.e., $O(\text{cost of the embedding layer} + \text{cost of other layers})$. The overall memory complexity of Phantom clipping remains the same as Ghost Clipping for all layers except the embedding layers. This advantage might diminish when the costs associated with other layers dominate the overall term. However, we note that when the model is small (suitable for local inference without the need for uploading sensitive data to the cloud service), Phantom clipping could achieve significant overall speedup by reducing the cost of the embedding layer.

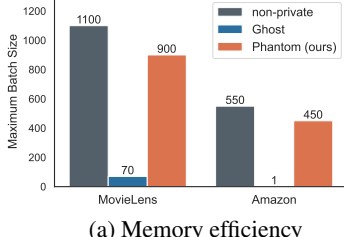
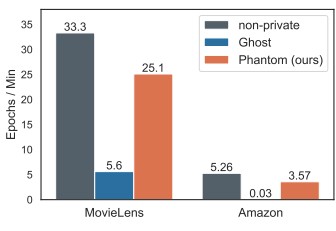

(a) Memory efficiency          (b) Training speed

Figure 3: Besides supporting parameter sharing, our phantom clipping provides additional speedup by leveraging the sparsity of input. **Left:** Phantom Clipping is 10-400× more memory efficient than Ghost Clipping and is almost as efficient as non-private training. **Right:** Phantom Clipping is 4-100× faster than Ghost Clipping, having comparable training speed with non-private training.

**Empirical Speedup.** We implement our Phantom Clipping based on AWS's fastDP[2] library, which has implemented Ghost Clipping. We then empirically compare our *Phantom Clipping* with Ghost Clipping in terms of both memory footprint and training speed on real-world datasets[3] (see Figure 5 for details of the datasets). Figure 3a shows the maximum batch size that can fit into a Tesla V100 GPU (16 GB of VRAM). It can be seen that our technique is much more memory friendly. It allows up to $450\times$ larger batch size compared with Ghost Clipping on Amazon, almost as large as those in

---

[2]https://github.com/awslabs/fast-differential-privacy

[3]Since Ghost Clipping does not support parameter sharing, its results are obtained from training models without embedding sharing. This leads to more model parameters. For a fair comparison, we halve its embedding dimension to $d_E/2$, ending up with a similar number of parameters as in the model with embedding sharing.

non-private training. Figure 3b shows the training speed on a single Tesla V100 GPU. It allows up to $100\times$ training speedup in practice compared to Ghost Clipping, achieving $0.68\times$ training speed of the non-private version.

# 4 RE-ATTENTION MECHANISM

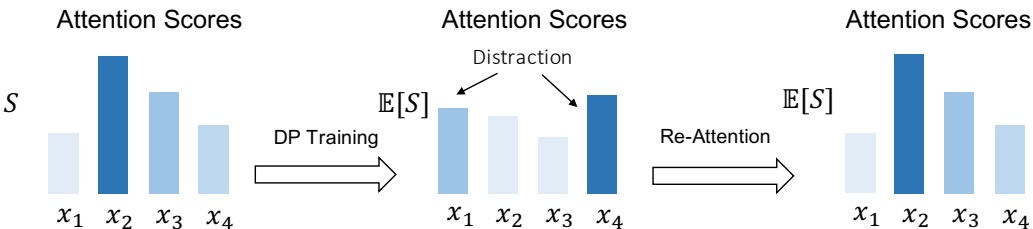

Figure 4: Illustration of Re-Attention Mechanism. Fix query $q$, consider attention scores with respect to $x_1$, $x_2$, $x_3$ and $x_4$, where $x_1$ and $x_4$ are assumed to be tail tokens. **Left:** The attention scores in non-private setting, i.e., ground truth, with the highest attention on tokens $x_2$ and $x_3$. **Middle:** Expectation of attention scores in DP training, where the attention is distracted to $x_4$ and $x_1$ due to the relatively higher uncertainty level of $x_1$ and $x_4$. **Right:** Re-Attention mechanism is designed to handle attention distraction by correcting the attention scores. See Appendix A.10 for full details.

## 4.1 MOTIVATION: THE ATTENTION DISTRACTION PHENOMENON

Recall that the Attention Mechanism, as the key component of the Transformer, given a query, calculates attention scores pertaining to tokens of relevance. Our key observation is that, over the randomness of the attention keys for the tokens of interest[4], the expectation of the attention scores will be distorted, particularly, mindlessly leaning towards tokens with high variance, regardless of their actual relevance. Refer to Figure 4 for a visual illustration of this concept of attention distraction.

To shed light on this phenomenon, we offer a theoretical analysis. Let us fix some query $q$ and denote the attention key of token $i$ as $K_i$. Since DP-SGD injects Gaussian noise into the model, it is natural to assume $K_i$ follows a Gaussian distribution with mean $k_i$ and variance $\sigma_i^2$. We denote $S_i$ as the random variable of the attention score assigned to token $i$. With some basic algebraic manipulation and applying the theory of extreme value (Coles et al., 2001), we can recast the formula for attention scores as follows[5],

$$S_i = \frac{\exp\langle q, K_i\rangle}{\sum_{j=1}^{L}\exp\langle q, K_j\rangle} = \exp\left(\langle q, K_i\rangle - \log\sum_{j=1}^{L}\exp\langle q, K_j\rangle\right) \tag{3}$$
$$= \exp\left(\langle q, K_i\rangle - \mathbb{E}_\gamma[\max_j\{\langle q, K_j\rangle + \gamma\}]\right),$$

where $\gamma$ is distributed as a standard Gumbel. Let us consider some token $i'$ that should have attracted little attention given the query $q$, then the expectation of the noisy maximum $\mathbb{E}_\gamma[\max_j\{\langle q, K_j\rangle + \gamma\}]$ can be approximated by $\max_{j\neq i'}\langle q, K_j\rangle + \zeta$, where $\zeta = \mathbb{E}[\gamma] = \pi^2/6$. Taking the expectation of Equation (3) over $K_{i'}$, and leveraging the fact $\mathbb{E}[\exp(X)] = \exp(\mathbb{E}[X])\exp(\mathrm{Var}[X]/2)$ when $X$ follows a Gaussian distribution, we then arrive at the following conclusion

$$\mathbb{E}_{K_{i'}}[S_{i'}] \approx \mathbb{E}_{K_{i'}}[\exp\left(\langle q, K_{i'}\rangle - (\max_{j\neq i'}\{\langle q, K_j\rangle\} + \zeta)\right)]$$
$$= \underbrace{\exp\left(\langle q, k_{i'}\rangle - \widetilde{M}\right)}_{\text{attentive relevance}} \cdot \underbrace{\exp\left(C\sigma_{i'}^2/2\right)}_{\text{multiplicative error}}, \tag{4}$$

where $\widetilde{M} = (\max_{j\neq i'}\{\langle q, K_j\rangle\} + \zeta)$ and the last equality leverages the fact that $\langle q, K_{i'}\rangle \sim \mathcal{N}(\langle q, k_{i'}\rangle, C\sigma^2)$ with $C = \langle q, q\rangle$. As a result, tokens with higher variance result in inflated attention scores due to increased multiplicative bias, distracting attention from more deserving tokens, given that token $i'$ is presupposed to garner little attention under query $q$. If all tokens have similar

---

[4]In unambiguous contexts, 'token variance' will denote the variance of the relevant representation associated with that token, like its attention key, $K_i$, or its embedding, $E_i$.

[5]For ease of notation, we omit the constant factor (i.e., $1/\sqrt{d}$) in attention computation.

variance or variance terms are negligible, the negative effects of this attention diversion are reduced. However, in less ideal conditions, especially with long-tailed data, this attention distraction could hinder the Transformer's training, thereby degrading model utility.

## 4.2 RE-ATTENTION VIA ERROR TRACKING

At its core, the Re-Attention Mechanism is designed to mitigate attention distraction during the learning process via debiasing the attention scores. To achieve this, it is natural to track the variance term identified as the error multiplier in Equation (26). In the following discussion, we elaborate on the methodology employed for tracking this error term during private training.

### 4.2.1 ERROR INSTANTIATION

Let us focus on the source of the randomness which leads to the attention distraction phenomenon, that is, the DP noise injected into the model gradients. Inspired by (Li et al., 2022b), we propose the idea of *effective error*, which is a probabilistic treatment of the *effective noise multiplier* in (Li et al., 2022b), proposed for model with sequential input. Effective error is used as an estimate of the uncertainty level underlying the model parameters, where the randomness is over DP noise.

**Definition 4.1. Effective Error:** The effective error $\sigma_{\text{eff}}^{\theta}$ associated with the model parameter $\theta$ is defined as

$$\sigma_{\text{eff}}^{\theta} = \frac{\sigma_{\text{dp}}}{B_{\text{eff}}^{\theta}}, \quad \text{where } B_{\text{eff}}^{\theta} = \mathbb{E}_{\mathcal{B} \overset{\text{i.i.d}}{\sim} \mathcal{D}^B} \left[ \sum_{i=1}^{B} \mathbb{I} \left[ R_{\theta}(\mathcal{B}_i) \right] \right], \quad (5)$$

where $B \in \mathbb{N}$ is the batch size, $\mathcal{B} \in \mathbb{N}^{B \times L}$ is the minibatch, i.i.d. sampled from training data distribution $\mathcal{D}$ (note that $\mathcal{B}_i$ is a sequence of tokens), $\sigma_{\text{dp}}$ is the DP noise multiplier in Equation (1), and $\mathbb{I}(\cdot)$ is the indicator function, $R_{\theta}(\cdot) = 1$ if $\mathcal{B}_i$ has relevance with $\theta$, for example, $R_{E_i}(\mathcal{B}_j) = \mathbb{I}[\text{token } i \in \mathcal{B}_j]$ where $E_i$ is the embedding for token $i$, and $R_W(\mathcal{B}_j) = 1$ where $W$ is the parameter within Transformer Block (see Figure 1).

**Remark 4.2.** *Effective error* recovers *effective noise multiplier* when the model has no embedding layer, for example, an MLP model. In that case, $\sigma_{\text{eff}}^{\theta} = \sigma_{\text{dp}}/B$.

We then have the following claims for obtaining effective error of the Transformer's parameters. See Appendix A.4 for detailed derivation.

**Claim 4.3.** *For each layer parameterized by $W$ within the Transformer block, its effective error is $\sigma_{\text{eff}}^{W} = \sigma_{\text{dp}}/B$.*

**Claim 4.4.** *For the embedding layer $E$, effective error of token $i$ is $\sigma_{\text{eff}}^{E_i} = \sigma_{\text{dp}}/(B \cdot p_i)$, where $p_i$ is the frequency of token $i$ (i.e., the probability of token $i$'s occurrence in data).*

### 4.2.2 ERROR PROPAGATION

Given the effective errors of the embedding layer and of the Transformer encoder, our goal is to obtain the error term $\sigma_i$ identified in Equation (26) for each attention computation. Notably, this issue of error tracking aligns with studies in Bayesian deep learning (Wang & Yeung, 2020), a field primarily focused on quantifying prediction uncertainty to enhance the robustness and reliability of machine learning systems. While our primary interest lies in unbiased attention score computation during private training, we can leverage and adapt existing methodologies in Bayesian deep learning to achieve this distinct goal. Specifically, given the input embedding along with its effective error, we propagate the effective error through Transformer layers (see Figure 1), with the goal of obtaining $\sigma_i$ for each attention calculation. We denote the output of the $l$-th layer by the random variable $X^{(l)}$. Given the output distribution $X^{(l-1)}$ of the preceding layer, the distribution $X^{(l)}$ can be computed layer-by-layer as follows,

$$p(X^{(l)}|X^{(0)}) = \mathbb{E}_{X^{(l-1)}|X^{(0)}} \left[ p\left( X^{(l)}|X^{(l-1)} \right) \right] = \mathbb{E}_{X^{(l-1)}|X^{(0)}} \left[ p^d \left( X_i^{(l)}|X^{(l-1)} \right) \right], \quad (6)$$

where the last equality is due to the isometric Gaussian noise of DP (see Equation 1), i.e., each dimension is independently and identically distributed. Based on Variational Inference (Kingma & Welling, 2013), we can use an approximating distribution $q$ to approximate the computationally intractable

distribution $p$, where $q(X^{(l)})$ follows a Gaussian distribution of mean $\mu$ and variance $\sigma^2$. Note that minimizing the KL divergence of $KL(p(X^{(l)}|X^{(0)})||q(X^{(l)}))$ reduces to matching the moments of $q(X^{(l)})$ to $p(X^{(l)}|X^{(0)})$. Since the mean and variance[6] are sufficient statistics for Gaussian distribution, propagating the distribution reduces to propagating its natural parameters (Wang et al., 2016). For linear layers coupled with a coordinate-wise non-linear activation, the statistics can be computed by analytic expressions using existing techniques from Probabilistic Neural Networks (Wang et al., 2016; Shekhovtsov & Flach, 2019; Gast & Roth, 2018; Postels et al., 2019; Morales-Alvarez et al., 2021). Concretely, for linear transformation, $X^{(l)} = X^{(l-1)}W$, we can propagate the variance as

$$\sigma^2_{X^{(l)}} = \sigma^2_{X^{(l-1)}} \cdot \sigma^2_W + \sigma^2_{X^{(l-1)}} \cdot \mu^2_W + \sigma^2_W \cdot (\mu_{X^{(l-1)}})^2. \tag{7}$$

For nonlinear activation functions, e.g., $X^{(l)} = \text{ReLU}\left(X^{(l-1)}\right)$, we can propagate the variance as

$$\sigma^2_{X^{(l)}} = \Phi(\frac{c}{\sqrt{d}})(c^2 + d) + \frac{c\sqrt{d}}{\sqrt{2\pi}}\exp(-\frac{1}{2}\frac{c^2}{d}) - c^2, \tag{8}$$

where $\Phi(\cdot)$ is the cumulative density function (CDF) of the standard Gaussian distribution, $c$ and $d$ are the natural parameter of $X^{(l-1)}$. For completeness, derivation is included in Appendix A.5.

All in all, we can obtain the output distribution of layer $(l)$ via analytic expression in terms of the natural parameter (Wang et al., 2016) of the preceding layer's output distribution as

$$(c^{(l)}, d^{(l)}) = \mathcal{F}(c^{(l-1)}, d^{(l-1)}), \quad \sigma^2 = \mathcal{T}(c^{(l)}, d^{(l)}). \tag{9}$$

Nevertheless, a nuanced difference exists between our error propagation and existing techniques encapsulated in Equation (9). In the Bayesian approach, the model parameter is directly associated with the mean $\mu_W$ in Equation (7). During private training, however, we can only access the noisy parameter after the injection of DP noise. Interestingly, access to this noisy parameter can be interpreted as a single sampling opportunity from its underlying Gaussian distribution, which can then be viewed as a one-time Markov Chain sampling (Wang et al., 2015). Therefore, the noisy parameter can serve as an estimate of its mean. In addition, unlike variance propagation in Bayesian deep learning, the error propagation here incurs minimal computational and memory overhead as the effective error can be represented in scalar (again, due to the isometric DP noise), plus the propagation is performed via analytical expressions.

**Re-Attention.** With the effective error tracked, we then proceed to mitigate the attention distraction identified in Equation (26) via $S_i \leftarrow S_i / \exp\left[C\sigma_i^2/2\right]$, obtaining unbiased attention scores.

In summary, we can propagate and track the effective error through the layers: given the natural parameter of $X^{(l-1)}$, the variance can be estimated using analytic expressions, which then can be used to correct the attention scores.

## 5 EXPERIMENTS

**Datasets and Prevalence of Long-Tailed Distributions.** We conduct experiments on two public recommendation datasets collected from real-world scenarios: Movie-Lens (Harper & Konstan, 2015) and Amazon (McAuley et al., 2015). Figure 5 shows their data distributions, illustrating the prevalence of long-tailed distributions, where a small number of items are extremely popular and have relatively high frequency while other items occur infrequently. The embedded table above the 'long tail' reports the statistics of the two datasets, showing that the two datasets vary significantly in size and sparsity. More details on datasets can be found in Appendix A.9.1.

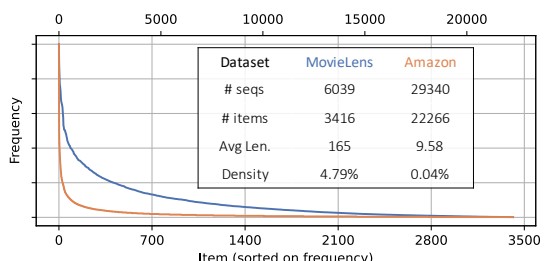

| Dataset | MovieLens | Amazon |
| --- | --- | --- |
| # seqs | 6039 | 29340 |
| # items | 3416 | 22266 |
| Avg Len. | 165 | 9.58 |
| Density | 4.79% | 0.04% |

Figure 5: Data in the real-world scenarios exhibits long-tailed (also known as, power-law) distribution.

---

[6]Note that for a Gaussian distribution, (i) mean and variance, (ii) the first two moments, and (iii) natural parameter, are equivalent in the sense of mutual convertibility. We will use them interchangeably.

**Baselines and Implementation Details.** We compare our DPFormer with vanilla Transformer (Vaswani et al., 2017) (i.e., the one without Re-Attention Mechanism), vanilla Transformer without parameter sharing, GRU (Cho et al., 2014), and LSTM (Hochreiter & Schmidhuber, 1997). For a fair comparison, embedding sharing is applied for all evaluated methods if not explicitly stated. The number of epochs is set to 100, where the first 20% of epochs are used for learning rate warm-up. After that, we linearly decay the learning rate through the remaining epochs. Following (Bu et al., 2022a; Yang et al., 2022), we normalize the gradients and set the clipping norm $C$ to 1, which eliminates the hyperparameter tuning for clipping norm $C$. For privacy accounting, we fix the total training epochs (iterations) and derive the noise required for each iteration from the preset privacy budget $\varepsilon$. More details on experiment setting can be found in Appendix A.9.

Table 1: Best results (%) on MovieLens at different privacy levels.

| DP Guarantee | $\varepsilon = 5$ | | $\varepsilon = 8$ | | $\varepsilon = 10$ | |
|---|---|---|---|---|---|---|
| Metric | NDCG@10 | HIT@10 | NDCG@10 | HIT@10 | NDCG@10 | HIT@10 |
| GRU | $2.26 \pm 0.04$ | $4.58 \pm 0.09$ | $2.40 \pm 0.03$ | $4.75 \pm 0.20$ | $2.81 \pm 0.03$ | $5.53 \pm 0.05$ |
| LSTM | $2.65 \pm 0.07$ | $5.08 \pm 0.08$ | $2.76 \pm 0.03$ | $5.41 \pm 0.06$ | $2.95 \pm 0.03$ | $5.55 \pm 0.06$ |
| TRANSFORMER W/O PS | $2.33 \pm 0.05$ | $4.47 \pm 0.07$ | $2.56 \pm 0.03$ | $5.11 \pm 0.05$ | $2.74 \pm 0.04$ | $5.39 \pm 0.08$ |
| TRANSFORMER (VANILLA) | $\mathbf{4.57 \pm 0.26}$ | $\mathbf{8.69 \pm 0.53}$ | $\mathbf{7.05 \pm 0.23}$ | $\mathbf{13.17 \pm 0.37}$ | $\mathbf{7.99 \pm 0.21}$ | $\mathbf{14.82 \pm 0.38}$ |
| DPFORMER (OURS) | $\mathbf{5.88 \pm 0.24}$ | $\mathbf{11.13 \pm 0.43}$ | $\mathbf{7.70 \pm 0.26}$ | $\mathbf{14.31 \pm 0.37}$ | $\mathbf{8.42 \pm 0.22}$ | $\mathbf{15.40 \pm 0.32}$ |
| Relative Improvement | 29%↑ | 28%↑ | 9.2%↑ | 8.7%↑ | 5.4%↑ | 3.9%↑ |

Table 2: Best results (%) on Amazon at different privacy levels.

| DP Guarantee | $\varepsilon = 5$ | | $\varepsilon = 8$ | | $\varepsilon = 10$ | |
|---|---|---|---|---|---|---|
| Metric | NDCG@10 | HIT@10 | NDCG@10 | HIT@10 | NDCG@10 | HIT@10 |
| GRU | $1.13 \pm 0.02$ | $2.46 \pm 0.03$ | $1.33 \pm 0.02$ | $2.22 \pm 0.02$ | $1.47 \pm 0.03$ | $2.48 \pm 0.02$ |
| LSTM | $1.19 \pm 0.01$ | $2.46 \pm 0.04$ | $1.23 \pm 0.01$ | $2.46 \pm 0.04$ | $1.34 \pm 0.01$ | $2.51 \pm 0.02$ |
| TRANSFORMER W/O PS | $1.16 \pm 0.01$ | $2.36 \pm 0.01$ | $1.20 \pm 0.02$ | $2.38 \pm 0.01$ | $1.40 \pm 0.01$ | $2.47 \pm 0.02$ |
| TRANSFORMER (VANILLA) | $\mathbf{1.37 \pm 0.04}$ | $\mathbf{2.47 \pm 0.10}$ | $\mathbf{1.54 \pm 0.03}$ | $\mathbf{2.77 \pm 0.07}$ | $\mathbf{1.57 \pm 0.03}$ | $\mathbf{2.83 \pm 0.08}$ |
| DPFORMER (OURS) | $\mathbf{1.64 \pm 0.01}$ | $\mathbf{3.01 \pm 0.01}$ | $\mathbf{1.98 \pm 0.05}$ | $\mathbf{3.70 \pm 0.15}$ | $\mathbf{1.99 \pm 0.04}$ | $\mathbf{3.73 \pm 0.11}$ |
| Relative Improvement | 20%↑ | 22%↑ | 28%↑ | 34%↑ | 27%↑ | 31%↑ |

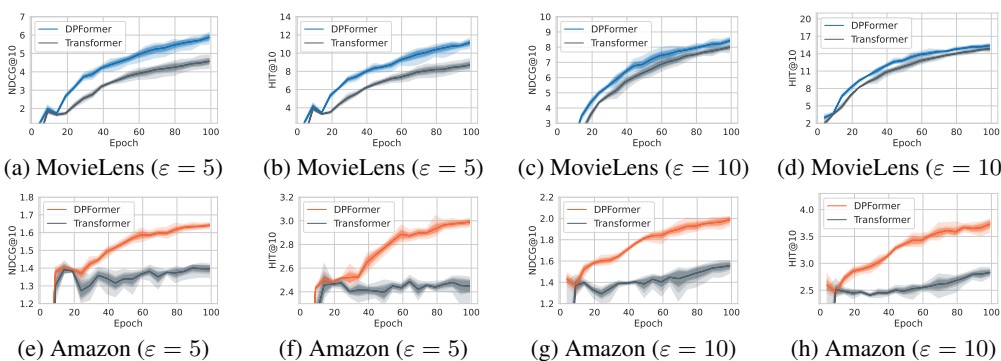

(a) MovieLens ($\varepsilon = 5$)    (b) MovieLens ($\varepsilon = 5$)    (c) MovieLens ($\varepsilon = 10$)    (d) MovieLens ($\varepsilon = 10$)

(e) Amazon ($\varepsilon = 5$)    (f) Amazon ($\varepsilon = 5$)    (g) Amazon ($\varepsilon = 10$)    (h) Amazon ($\varepsilon = 10$)

Figure 6: The Re-Attention Mechanism renders DPFormer notably more stable during private training. Each run is repeated five times with independent random seeds, with test accuracy (i.e., NDCG@10(%) and HIT@10(%)) reported every five epochs. The graduated shading (best viewed zoomed in) represents confidence intervals from 60% to 100%.

Table 1 and Table 2 show the best[7] NDCG@10 and HIT@10 for all the methods on MovieLens and Amazon. The vanilla Transformer outperforms all other baselines, reaffirming its dominance

---

[7] Strictly speaking, the process of hyperparameter tuning would cost privacy budget (Papernot & Steinke, 2022), but is mainly of theoretical interest. We perform grid search on learning rate $\in \{10^{-3}, 3 \times 10^{-3}, 5 \times 10^{-3}, 7 \times 10^{-3}, 9 \times 10^{-3}\}$ and batch size $\in \{256, 512, 1024, 2048, 4096\}$ for each method, ensuring fair comparison.

in sequential data modeling due to the Attention Mechanism. Our DPFormer, incorporating the Re-Attention Mechanism, further boosts the performance by around 20% on average. Notably, under a low privacy budget ($\varepsilon = 3$), DPFormer achieves a relative improvement of around 25%, demonstrating its efficacy in attenuating attention distraction during private training. On MovieLens, as expected, the performance gain increases with decreasing privacy budget $\varepsilon$, i.e., increasing noise strength during training; this is because larger noise corresponds to more severe attention distraction, which better highlights the Re-Attention Mechanism's advantage. However, on Amazon, DPFormer achieves a smaller relative improvement at $\varepsilon = 5$ than at $\varepsilon = 10$. We suspect that this is due to the two datasets' differences in terms of sparsity (i.e., $1-$density in Figure 5) as well as the inherent hardness of training Transformer (Zhang et al., 2019; Xu et al., 2020; Huang et al., 2020) and substantial DP noise.

Figure 6 shows the model accuracy every five epochs during training. Evidently, the training dynamics of the vanilla Transformer, impacted by attention distraction, can suffer from high variance and/or substantial fluctuation, especially on Amazon. In contrast, DPFormer enjoys faster and smoother convergence, highlighting its superior training stability under differential privacy.

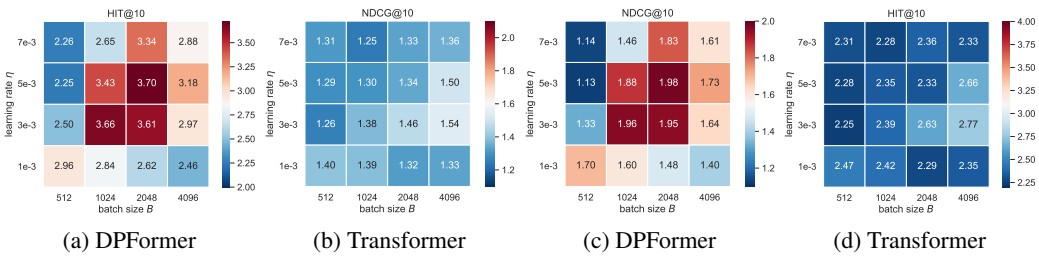

| (a) DPFormer | (b) Transformer | (c) DPFormer | (d) Transformer |

Figure 7: Results of grid search for hyperparameter tuning on Amazon with privacy budget $\varepsilon = 8$.

To study the robustness and sensitivity with respect to the hyperparameters of our method, Figure 7 shows the results of hyperparameter tuning via grid search [8]. For *reasonable* (along the main diagonal (Tramer & Boneh, 2021)) hyperparameter configurations, our DPFormer significantly and consistently outperforms the vanilla Transformer.

## 6  CONCLUSION

In this paper, we identify two key challenges in learning differentially private Transformers, i.e., heavy computation overhead due to per-sample gradient clipping and attention distraction due to long-tailed data distributions. We then proposed DPFormer, equipped with Phantom Clipping and Re-Attention Mechanism, to address these challenges. Our theoretical analysis shows that DPFormer can effectively correct attention shift (which leads to significant performance drops) and reduce computational cost during gradient clipping, which is further corroborated by empirical results on two real-world datasets with varying degrees of long-tailedness. We hope our work has the potential to spur future research.

**Limitation.** Firstly, while the relative performance gain is significant when privacy budget is relative low ($\varepsilon = 5$), the performance still suffers, which drops sharply when $\varepsilon$ decreases from 8 to 5, in contrast to the more moderate decline observed when $\varepsilon$ reduces from 10 to 8. It remains unclear whether we have already achieved the limit of private learning (where the hardness is mainly posed by the limited and long-tailed data) or there is still potential for improvement. Secondly, we do not scale our method to the large models in this work. While the use of small model is well justified considering the computing resource of end devices (which is preferable for privacy protection through local inference), it remains to be seen to which degree the improvement can be obtained by the Phantom Clipping and Re-Attention Mechanism in terms of efficiency and effectiveness when adopting a larger Transformer model. Exploring these directions would be interesting future work.

---

[8]Rather than run an addtional differentially private algorithm to report a noisy max (or argmax) (Papernot & Steinke, 2022), we opt for this practice of directly displaying all results due to its transparency and comprehensiveness.

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

# A APPENDIX

## A.1 RELATED WORK

### A.1.1 DIFFERENTIAL PRIVATE DEEP LEARNING

Papernot et al. (2021) suggested tempered sigmoid activations to control the gradient norm explicitly, and in turn support faster convergence in the settings of differentially private ML. Mohapatra et al. (2022) studied the intrinsic connection between the learning rate and clipping norm hyperparameters and show that adaptive optimizers like DPAdam enjoy a significant advantage in the process of honest hyperparameter tuning. Wei et al. (2022) employed importance sampling (IS) in each SGD iteration for minibatch selection.

One line of work focuses on adaptive optimization of differentially private machine learning. Asi et al. (2021) proposed adaptive stepsizes as a variant of SGD. Li et al. (2022a) uses non-sensitive side information to precondition the gradients, allowing the effective use of adaptive methods in private settings.

Another line of work studies the loss landscape of DP-SGD in comparison to SGD. Wang et al. (2021) first highlighted the problem of DP-SGD being stuck in local minima due to the training instability. They constructed a smooth loss function that favors noise-resilient models lying in large flat regions of the loss landscape. Shamsabadi & Papernot (2021) proposed that loss functions with smaller norm can reduce the impact of clipping and thus create a smoother loss function. Park et al. (2023) makes use of sharpnessaware training without additional privacy costs.

The most related work is Li et al. (2022b), which finetunes large language models with differential privacy. They propose Ghost Clipping, which is a technique that enables efficient per-sample gradient clipping without instantiating per-sample gradient. Our Phantom Clipping can be viewed as an extension of Ghost Clipping that additionally handles parameter sharing of the embedding layer. They also introduce the idea of *effective noise multiplier* in order to explain the role of batch size in private learning. Our *effective error* (Equation 5) can be viewed as its generalization in order to account for the inherent input sparsity (i.e., only a small portion of tokens appear in one training sequence).

Another related work is Anil et al. (2022), which establishes a baseline for BERT-Large pretraining with DP. They introduce several strategies to help private training, such as large weight decay and increasing batch size schedule. Notably, these strategies are independent yet complementary to the methodologies utilized in this work, thereby offering potential avenues for an integrated approach.

### A.1.2 LEARNING ON LONG-TAILED DATA

Previous work has also considered the differentially private learning algorithms on heavy-tailed data Wang et al. (2020); Hu et al. (2022); Kamath et al. (2022). This line of research is mainly concerned with differential private stochastic optimization (DP-SCO). Note that the notion of heavy-tailed there is different from the focus of this work. As pointed out in Kamath et al. (2022), the setting they actually consider is dealing with heavy-tailed gradients due to unbounded values in the input data.

### A.1.3 FAIRNESS IN PRIVATE LEARNING ON LONG-TAILED DATA

Sanyal et al. (2022) observed that when the data has a long-tailed structure, it is not possible to build accurate learning algorithms that are both private and results in higher accuracy on minority subpopulations. The goal is to obtain higher minority group accuracy, albeit at the cost of the overall accuracy. In contrast, our emphasize is on obtaining higher overall accuracy in the presence of long-tailed data.

## A.2 PROOF OF PHANTOM CLIPPING (CLAIM 16)

Let $B$ be the batch size. Let $L \in \mathbb{N}$ be the length of each sequence. Let $s_i \in \mathbb{N}^L$ be a training sample/sequence in a minibatch. Let $M \in \mathbb{N}$ be the vocabulary size. Let $E \in \mathbb{R}^{M \times d}$ be the (shared) embedding layer.

Let $a_{i,s} \in \{0,1\}^{L \times M}$ be the one-hot encodings of the sequence $s_i$, i.e., the input of the input embedding layer.

Let $e_{i,s} \in \mathbb{R}^{L \times d}$ be output of the input embedding layer when fed into $a_{i,s}$,

$$e_{i,s} = \text{InputEmbedding}_E(a_{i,s}). \tag{10}$$

Let $\tilde{e}_i \in \mathbb{R}^d$ be the output of the transformer encoder when fed into $e_{i,s}$

Let $a_{i,c} \in \{0,1\}^{M \times M}$ be the input of the output embedding layer. Let $e_{i,c} \in \mathbb{R}^{M \times d}$ be output of the output embedding layer when fed into $a_{i,c}$,

$$e_{i,s} = \text{OutputEmbedding}_E(a_{i,c}). \tag{11}$$

The prediction score $r_i \in \mathbb{R}^M$ for the next token of the sentence $s_i$ is obtained by

$$r_i = e_{i,s} \cdot \tilde{e}_i. \tag{12}$$

Let $\mathcal{L} = \sum_i^B \mathcal{L}_i$ be the total loss where $\mathcal{L}_i$ is the per-sample loss with respect to the $i$-th sample,

$$\mathcal{L}_i = \text{CrossEntropy}(r_i, y_i), \tag{13}$$

where $y_i$ is the ground truth, i.e., the next token of $s_i$.

During a standard back-propagation, we can obtain

$$\nabla e_{i,s} := \partial \mathcal{L}_i / \partial e_{i,s} \in \mathbb{R}^{L \times d}, \tag{14}$$

$$\nabla e_{i,c} := \partial \mathcal{L}_i / \partial e_{i,c} \in \mathbb{R}^{M \times d}. \tag{15}$$

**Claim A.1.** *(Phantom Clipping) The norm of the per-sample gradient with respect to $E$ can be efficiently evaluated as*

$$\|g_{i,E}\| = \left( \langle a_{i,s} a_{i,s}^T, \nabla e_{i,s} \nabla e_{i,s}^T \rangle^2 + \|\nabla e_{i,c}\|^2 + 2 \cdot \langle \nabla e_{i,s}, a_{i,s}^T \nabla e_{i,c} \rangle \right)^{\frac{1}{2}}, \tag{16}$$

*where $\langle \cdot, \cdot \rangle$ is the inner product of two matrices being of the same shape.*

*Proof.* For simplicity, we will omit the per-sample index $i$ throughout this proof. From the chain rule, the per-sample gradient with respect to the embedding layer $E$ is

$$g_E = \frac{\partial \mathcal{L}}{\partial e_s} \cdot \frac{\partial e_s}{\partial E} + \frac{\partial \mathcal{L}}{\partial e_c} \cdot \frac{\partial e_c}{\partial E}$$
$$= \underbrace{a_s^T \cdot \nabla e_s}_{g_E^{(1)}} + \underbrace{a_c^T \cdot \nabla e_c}_{g_E^{(2)}}, \tag{17}$$

where $a_s \in \{0,1\}^{L \times M}$ (or $a_c \in \{0,1\}^{M \times M}$) is the one-hot encodings of the input sequence $s_i$ (or those of the candidate tokens for the output probability) in a minibatch, and $e_s \in \mathbb{R}^{L \times d}$ (or $e_c \in \mathbb{R}^{M \times d}$) be output of the (shared) embedding layer $E$ when fed into $a_s$ (or $a_c$). Denote the first segment of the right-hand side (RHS) as $g_E^{(1)}$, the second segment of the RHS as $g_E^{(2)}$. Then we have

$$\|g_E\|_F^2 = \left\|g_E^{(1)} + g_E^{(2)}\right\|_F^2 = \left\|g_E^{(1)}\right\|_F^2 + \left\|g_E^{(2)}\right\|_F^2 + 2 \cdot \langle g_E^{(1)}, g_E^{(2)} \rangle. \tag{18}$$

Ghost Clipping (Li et al., 2022b) allows us to evaluate $\left\|g_E^{(1)}\right\|_F$ without instantiating $g_E^{(1)}$, the formula is given by

$$\left\|g_E^{(1)}\right\|_F = \langle a_s a_s^T, \nabla e_s \nabla e_s^T \rangle. \tag{19}$$

Similarly, we have

$$\left\|g_E^{(2)}\right\|_F = \langle a_c a_c^T, \nabla e_c \nabla e_c^T \rangle. \tag{20}$$

Note that $a_{i,c}$ is the one-hot encoding of $[1, 2, 3, ..., M]$, thus $a_{i,c}$ is an Identity matrix, Equation 20 can be further simplified as

$$\left\| g_{i,E}^{(2)} \right\|_F = \langle \mathbf{I}, \nabla e_{i,c} \nabla e_{i,c}^T \rangle = \sum_{i=1}^{M} \langle (\nabla e_s)_i, (\nabla e_s)_i \rangle = \| \nabla e_{i,c} \|^2. \tag{21}$$

Note that this simplification reduces the memory footprint from $O(M^2)$ to $O(M)$, obviating the need for evaluating $e_c \nabla e_c^T$ in Equation 20.

Therefore, computing the gradient norm of shared embedding reduces to computing $\langle g_E^{(1)}, g_E^{(2)} \rangle$ in Equation 18.

$$\begin{aligned}
\langle g_E^{(1)}, g_E^{(2)} \rangle &= \langle a_s^T \cdot \nabla e_s, a_c^T \cdot \nabla e_c \rangle \\
&= \sum_{j=1}^{M} \sum_{k=1}^{d} \left( \sum_{i=1}^{L} (a_s)_{ij} \cdot (\nabla e_s)_{ik} \right) \left( \sum_{i=1}^{M} (a_c)_{ij} \cdot (\nabla e_c)_{ik} \right) \\
&= \sum_{i_1=1}^{L} \sum_{i_2=1}^{M} \left( \sum_{j=1}^{M} \sum_{k=1}^{d} (a_s)_{i_1 j} \cdot (\nabla e_s)_{i_1 k} \cdot (a_c)_{i_2 j} \cdot (\nabla e_c)_{i_2 k} \right) \\
&= \sum_{i_1=1}^{L} \sum_{i_2=1}^{M} \left( \sum_{j=1}^{M} (a_s)_{i_1 j} \cdot (a_c)_{i_2 j} \right) \left( \sum_{k=1}^{d} (\nabla e_s)_{i_1 k} \cdot (\nabla e_c)_{i_2 k} \right) \\
&= \sum_{i_1=1}^{L} \sum_{i_2=1}^{M} \langle (a_s)_{i_1}, (a_c)_{i_2} \rangle \cdot \langle (\nabla e_s)_{i_1}, (\nabla e_c)_{i_2} \rangle \\
&= \sum_{i_1=1}^{L} \sum_{i_2=1}^{M} [i_2 == \text{onehot}^{-1}((a_s)_{i_1})] \cdot \langle (\nabla e_s)_{i_1}, (\nabla e_c)_{i_2} \rangle \\
&= \sum_{i_1=1}^{L} \langle (\nabla e_s)_{i_1}, (a_s^T \cdot \nabla e_c)_{i_2} \rangle \\
&= \langle (\nabla e_s), a_s^T \cdot (\nabla e_c) \rangle.
\end{aligned} \tag{22}$$

Combining Equation 19, 21 and 22 yields the conclusion. $\qquad \square$

## A.3 MOTIVATION OF RE-ATTENTION MECHANISM

Each each iteration of model training needs randomness (e.g. sample a mini-batch, add DP noise). We split the randomness into two disjoint parts, the coin flipping $r_1 \in \{0, 1\}^*$ is inherited from non-private machine learning, the coin flipping $r_2 \in \{0, 1\}^*$ is additionally required for DP noise. We fix $r_1$ for now and let $\boldsymbol{r_2}$ be the uniform random variable over $\{0, 1\}^*$.

Conditioned on the model parameter (before adding DP noise) at the end of the iteration $t - 1$ is $\theta_{(t-1)}^{\text{non\_private}}$. By injecting DP noise, we have

$$\boldsymbol{\theta_{(t-1)}} = \text{Privatize}(\theta_{(t-1)}^{\text{non\_private}}, \boldsymbol{r_2}), \tag{23}$$

where $\text{Privatize}(\cdot)$ refers to adding Gaussian noise to the model parameter for differential privacy, which takes as input the non-private version of the model parameter $\theta_{(t-1)}^{\text{non\_private}}$ and the randomness $r_2$ required for sampling Gaussian noise. The distribution of random variable $\boldsymbol{\theta_{(t-1)}}$ is induced by the uniform random variable $\boldsymbol{r_2}$. Namely, $\boldsymbol{\theta_{(t-1)}}$ follows a Gaussian distribution with mean $\theta_{(t-1)}^{\text{non\_private}}$ and variance $\sigma_{DP}$. It is important to stress that we are not allowed to access $\theta_{(t-1)}^{\text{non\_private}}$, otherwise this could lead to the violation of differential privacy. Instead, what we can access freely is the noisy parameter $\theta_{(t-1)}^{\text{private}}$ after noise injection. Note that access to this noisy parameter can be interpreted as a single sampling opportunity from its underlying Gaussian distribution $\boldsymbol{\theta_{(t-1)}}$, which can then be viewed as a one-time Markov Chain sampling (Wang et al., 2015). Therefore, the noisy parameter

$\theta_{(t-1)}^{\text{private}}$ can serve as an estimate of its mean $\theta_{(t-1)}^{\text{non\_private}}$. Jumping ahead, we will exploit this fact in our Re-Attention method.

Without loss of generality, assume the batch size is 1. At the beginning of the iteration $t$, we sample a training sequence, denoted by $seq \in \mathbb{N}^L$ where $L \in \mathbb{N}$ is the length of the sequence, using $r_1$ as the randomness and feed $seq$ into the model for forward propagation. Before performing the attention score calculation, it will first obtain the key and query as follows,

$$\boldsymbol{K}, \boldsymbol{q} = \text{PreAttention}(\boldsymbol{\theta_{(t-1)}}, \mathcal{D}, r_1), \tag{24}$$

where $\mathcal{D}$ is the training data set, $\boldsymbol{K} \in \mathbb{R}^{d \times L}$ ($d \in \mathbb{N}$ is the model dimension) is the key (required for attention score calculation) random variable whose distribution is induced by $\boldsymbol{\theta_{(t-1)}}$. Likewise, $\boldsymbol{q} \in \mathbb{R}^d$ is the query random variable.

We fix a query $q \in \text{Supp}(\boldsymbol{q})$. For $i \in [L]$, let $S_i \in \mathbb{R}$ be the attention score of $i$-th token in $seq$ calculated from $q$ and $\boldsymbol{K}$. With some basic algebraic manipulation and applying the theory of extreme value (Coles et al., 2001), we can recast the formula for attention scores as follows[9],

$$
\begin{aligned}
\boldsymbol{S}_i &= \text{Attention}(q, \boldsymbol{K}_i) \\
&= \frac{\exp \langle q, \boldsymbol{K}_i \rangle}{\sum_{j=1}^{L} \exp \langle q, \boldsymbol{K}_j \rangle} \\
&= \exp \left( \langle q, \boldsymbol{K}_i \rangle - \log \sum_{j=1}^{L} \exp \langle q, \boldsymbol{K}_j \rangle \right) \\
&= \exp \left( \langle q, \boldsymbol{K}_i \rangle - \mathbb{E}_{\boldsymbol{\gamma}}[\max_j\{\langle q, \boldsymbol{K}_j \rangle + \boldsymbol{\gamma}\}] \right),
\end{aligned} \tag{25}
$$

where $\boldsymbol{\gamma}$ is distributed as a standard Gumbel. Let us consider some token $i' \in [L]$ that should have attracted little attention given the query $q$, then the expectation of the noisy maximum $\mathbb{E}_{\boldsymbol{\gamma}}[\max_j\{\langle q, \boldsymbol{K}_j \rangle + \boldsymbol{\gamma}\}]$ can be approximated by $\max_{j \neq i'} \langle q, \boldsymbol{K}_j \rangle + \zeta$, where $\zeta = \mathbb{E}[\gamma] = \pi^2/6$. Taking the expectation of Equation (3) over $K_{i'}$, and leveraging the fact $\mathbb{E}[\exp(X)] = \exp(\mathbb{E}[X]) \exp(\text{Var}[X]/2)$ when $X$ follows a Gaussian distribution, we then arrive at the following conclusion

$$
\begin{aligned}
\mathbb{E}_{K_{i'}}[\boldsymbol{S_{i'}}] &\approx \mathbb{E}_{K_{i'}}[\exp(\langle q, \boldsymbol{K}_{i'} \rangle - (\max_{j \neq i'}\{\langle q, \boldsymbol{K}_j \rangle\} + \zeta))] \\
&= \underbrace{\exp(\langle q, \boldsymbol{K}_{i'} \rangle - \widetilde{M})}_{\text{attentive relevance}} \cdot \underbrace{\exp\left(C\sigma_{i'}^2/2\right)}_{\text{multiplicative error}},
\end{aligned} \tag{26}
$$

where $\widetilde{M} = (\max_{j \neq i'}\{\langle q, K_j \rangle\} + \zeta)$ and the last equality leverages the fact that $\langle q, K_{i'} \rangle \sim \mathcal{N}(\langle q, k_{i'} \rangle, C\sigma^2)$ with $C = \langle q, q \rangle$. As a result, tokens with higher variance result in inflated attention scores due to increased multiplicative bias, distracting attention from more deserving tokens, given that token $i'$ is presupposed to garner little attention under query $q$. If all tokens have similar variance or variance terms are negligible, the negative effects of this attention diversion are reduced. However, in less ideal conditions, especially with long-tailed data, this attention distraction could hinder the Transformer's training, thereby degrading model utility.

## A.4 PROOF OF ERROR INSTANTIATION (CLAIM 4.3 AND 4.4)

Taking the average over the minibatch in Equation 1 can be considered noise reduction of $O(1/B)$. Fix the noise multiplier $\sigma_{dp}$. As the size of the batch increases, the amount of DP noise incorporated into the parameter decreases correspondingly. Suppose that token $i$ is absent from the current training sequence. Its input embedding will not be activated and thus will not be properly trained in this iteration, but the DP noise will be nevertheless injected into its embedding. The concept of effective error is introduced to account for this phenomenon.

*Proof.* It reduces the derive the formula for effective batch size $B_{\text{eff}}^\theta$ in Equation 5. Recall that its definition is given by

$$B_{\text{eff}}^\theta = \mathbb{E}_{\mathcal{B} \overset{\text{i.i.d}}{\sim} \mathcal{D}^B} \left[ \sum_{i=1}^{B} \mathbb{I}\left[R_\theta(\mathcal{B}_i)\right] \right]. \tag{27}$$

---

[9]For ease of notation, we omit the constant factor (i.e., $1/\sqrt{d}$) in attention computation.

For each layer parameterized by $W$ within the Transformer block, its effective batch size is $B_{\text{eff}}^W = B$, since $R_W(\mathcal{B}_i) = 1$.

For the embedding layer $E$, its effective batch size is

$$
\begin{aligned}
B_{\text{eff}}^{E_i} &= \mathbb{E}_{\mathcal{B} \overset{\text{i.i.d}}{\sim} \mathcal{D}^B} \left[ \sum_{j=1}^{B} \mathbb{I} \left[ R_{E_i}(\mathcal{B}_j) \right] \right] \\
&= \sum_{j=1}^{B} \mathbb{E}_{\mathcal{B}_j \overset{\text{i.i.d}}{\sim} \mathcal{D}} \left[ \mathbb{I} \left[ R_{E_i}(\mathcal{B}_j) \right] \right] \quad \text{(Linearity of Expectation)} \\
&= \sum_{j=1}^{B} \mathbb{E}_{\mathcal{B}_j \overset{\text{i.i.d}}{\sim} \mathcal{D}} \left[ \mathbb{I} \left[ \text{token } i \in \mathcal{B}_j \right] \right] \\
&= \sum_{j=1}^{B} p_i \\
&= B \cdot p_i,
\end{aligned}
\tag{28}
$$

where $p_i$ is the frequency of token $i$ (i.e., the probability of token $i$'s occurrence in data). $\qquad \square$

**Remark A.2.** Note that $p_i$, as the frequency statistics, can be obtained with high accuracy with tiny privacy budget. For simplicity, we will assume $p_i$ is publicly known in this work. We note that this is a realistic assumption because the total number of clicks/purchase of each item is public in many online commercial platform powered by recommendation systems.

### A.5 PROOF OF LINEAR TRANSFORMATION (EQUATION 7)

**Lemma A.3.** *Let $X$, $Y$ be two independent random variables. Let $Z = XY$, then the variance of $Z$ can be expressed as*

$$
\mathrm{Var}[Z] = \mathrm{Var}[XY] = \mathbb{E}[X^2]\mathbb{E}[Y^2] - \mathbb{E}[XY]^2.
\tag{29}
$$

LemmaA.3 directly implies Equation 7 as follows.

*Proof.* Suppose the linear transformation is given by $X^{(l)} = X^{(l-1)}W$, then we have

$$
\begin{aligned}
\mathrm{Var}[X^{(l)}] &= \mathbb{E}[(X^{(l-1)})^2]\mathbb{E}[W^2] - \mathbb{E}[X^{(l-1)}W]^2 \\
&= (\mathbb{E}[X^{(l-1)}]^2 + \mathrm{Var}[X^{(l-1)}])(\mathbb{E}[W]^2 + \mathrm{Var}[W]) - \mathbb{E}[X^{(l-1)}]^2\mathbb{E}[W]^2 \\
&= \mathrm{Var}[X^{(l-1)}]\mathrm{Var}[W] + \mathbb{E}[X^{(l-1)}]^2\mathrm{Var}[W] + \mathbb{E}[W]^2\mathrm{Var}[X^{(l-1)}].
\end{aligned}
\tag{30}
$$

$\qquad \square$

### A.6 PROOF OF NON-LINEAR TRANSFORMATION (EQUATION 9)

**Lemma A.4.** *Let $X_1$, $X_2$ be two independent Gaussian random variables, where $X_i \sim \mathcal{N}(\mu_i, \sigma_i), i = 1, 2$. Let $Z = \max(X_1, X_2)$.*

$$
\begin{aligned}
\mathbb{E}[Z] &= \mu_1 \Phi(\gamma) + \mu_2 \Phi(-\gamma) + \nu \phi(\gamma) \\
\mathbb{E}[Z^2] &= (\mu_1^2 + \sigma_1^2)\Phi(\gamma) + (\mu_2^2 + \sigma_2^2)\Phi(-\gamma) + (\mu_1 + \mu_2)\nu\phi(\gamma),
\end{aligned}
\tag{31}
$$

*where $\Phi(\cdot)$ is the cumulative density function (CDF) of the standard Gaussian distribution, $\phi(\cdot)$ is the probability density function (PDF) of the standard Gaussian distribution, $\nu = \sqrt{\sigma_1^2 + \sigma_2^2}$, and $\gamma = (\mu_1 - \mu_2)/\nu$.*

LemmaA.4 directly implies Equation 9 as follows.

*Proof.* Let $X^{(l)} = \mathrm{ReLU}\left(X^{(l-1)}\right)$ be the ReLU activation. Substitute $\mu_1 = \mathbb{E}[X^{(l-1)}], \sigma_2 = \mathrm{Var}[X^{(l-1)}], \mu_2 = \sigma_2 = 0$ into Equation 31. Leveraging $\mathrm{Var}[X^{(l)}] = \mathbb{E}[(X^{(l)})^2] - \mathbb{E}[X^{(l)}]^2$ yields the conclusion. $\square$

**Remark A.5. GELU activation.** GELU function is another widely used activation function within Transformer models. GELU can be viewed as a smooth version of ReLU (see Figure 8), where their forward propagation is similar, and the major distinction lies in the numerical behavior of backpropagation. Since error propagation is only concerned with forward propagation behavior, we can also use Equation 9 to approximate the variance of GELU output. Table 3 shows the analytic error propagation for ReLU and GELU activation, compared with the sampling-based results.

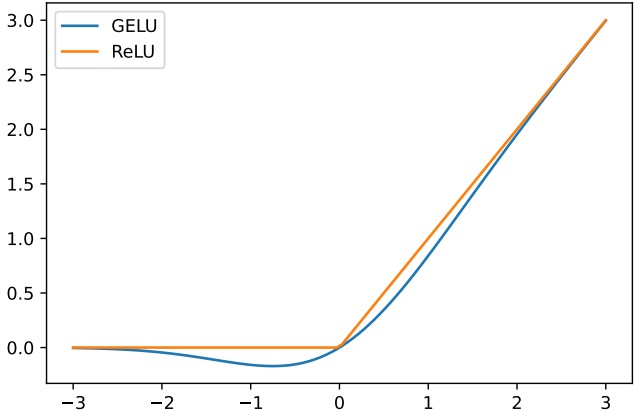

Figure 8: GELU activation and ReLU activation.

Table 3: Analytic error propagation for ReLU and GELU activation.

| Input | Activation | Sampling-based | | | | | | Analytic |
|---|---|---|---|---|---|---|---|---|
| | | 10 | 100 | 1000 | 10000 | 100000 | 1000000 | |
| $\mathcal{N}(0, 0.01)$ | ReLU | 4.08e-6 | 3.60e-5 | 3.93e-5 | 3.45e-5 | 3.40e-5 | 3.40e-5 | **3.40e-5** |
| | GELU | 2.48e-5 | 2.69e-5 | 2.72e-5 | 2.57e-5 | 2.50e-0 | 2.49e-5 | |
| $\mathcal{N}(0, 0.1)$ | ReLU | 0.0030 | 0.0031 | 0.0037 | 0.0034 | 0.0035 | 0.0034 | **0.0034** |
| | GELU | 0.0030 | 0.0025 | 0.0027 | 0.0025 | 0.0026 | 0.0025 | |
| $\mathcal{N}(0, 1)$ | ReLU | 0.5299 | 0.2361 | 0.3649 | 0.3451 | 0.3387 | 0.3418 | **0.3408** |
| | GELU | 0.5525 | 0.2306 | 0.3719 | 0.3506 | 0.3433 | 0.3467 | |

## A.7 RE-ATTENTION MECHANISM

The algorithm description of the proposed Re-Attention Mechanism is presented in Algorithm 1.

## A.8 PRIVACY GUARANTEE

**Claim A.6.** *Given the noise multiplier, batch size, number of samples, training epochs, let $(\varepsilon, \delta)$ be the derived parameter for DP guarantee by the privacy accountant of DP-SGD. DPFormer is $(\varepsilon, \delta)$-DP.*

*Proof.* The privacy guarantee is inherited from that of DP-SGD. Details follow.

Phantom clipping is an efficient implementation of gradient clipping, having the same functionality as vanilla per-sample clipping (i.e., the same input-output behavior). Therefore, we can think of it as the vanilla gradient clipping when we analyze the privacy.

For Re-Attention mechanism, observe that the whole computation process only depends on the single data sample (i.e., we do not create dependency among samples between a mini-batch, thus violating

---

**Algorithm 1** Re-Attention Mechanism

---

**Input**: $X \in \mathbb{N}^{B \times L}$: A minibatch of training sentences; $B$: batch size; $L$: sentence length.
**Parameter**: Privacy parameters $\varepsilon, \epsilon > 0, \delta \in (0, 1)$;

1: Calculate $\sigma_{\text{DP}} \leftarrow$ PrivacyAccountant
2: $c^{(0)}, d^{(0)} =$ ErrorInstantiation;                                    ▷ Section 4.2.1
3: Initialize $X^{(0)} = X$;
4: **for** $l$ in [1, 2, ..., #layers of NN] **do**
5:     $X^{(l+1)} =$ PreAttenionFeedForward($X^{(l)}$);
6:     $S \leftarrow$ Self-Attention($X^{(l+1)}$)                             ▷ The vanilla self-attention
7:     $\sigma =$ Convert($c^{(l)}, d^{(l)})$)
8:     $S \leftarrow S/\exp\left[C\sigma^2/2\right]$                          ▷ Re-Attention mechanism
9:     $X^{(l+1)} =$ PostAttenionFeedForward($S_i$);
10:    $c^{(l+1)}, d^{(l+1)} =$ ErrorPropagation($c^{(l)}, d^{(l)}$);          ▷ Section 4.2.2

---

DP like Batch Normalization). Another way to understand this is we can do whatever we like in the forward pass as long as the computation associated with a single input sample does not rely on other input samples of the mini-batch. Note that a caveat is discussed in Remark A.1. of Appendix. Therefore, the private analysis reduces to DP-SGD: By clipping the gradient we can successfully bound the per-sample sensitivity (and add the calibrated noise to guarantee DP).                    □

### A.9 EXPERIMENTAL DETAILS

#### A.9.1 DATASETS

**MovieLens.** The MovieLens dataset (Harper & Konstan, 2015) is often used in the development and evaluation of collaborative filtering algorithms, which are used to make personalized recommendations based on user behavior. It is a benchmark dataset in the field of recommender systems due to its size, longevity, and richness of user-item interactions. We use the version (MovieLens-1M) that includes 1 million user behaviors.

**Amazon.** A series of datasets introduced in (McAuley et al., 2015), comprising large corpora of product reviews crawled from Amazon.com. Top-level product categories on Amazon are treated as separate datasets. We consider the 'Games.' category. This dataset is notable for its high sparsity and variability.

We follow (Kang & McAuley, 2018) for the data preprocessing. We use timestamps to determine the sequence order of actions. Each user is associated with a training sequence (i.e., his chronological behavior). We discard users and items with fewer than five related actions. For data partitioning, the last token of each sequence is left for testing.

It is worth noting that since each user is exclusively associated with exactly one training sample (sequence) in the training data, the DP guarantee we provide is user-level. That is, removing all information pertaining to a specific user yields an indistinguishable model.

#### A.9.2 MODEL ARCHITECTURE

We use the standard Transformer encoder described in (Vaswani et al., 2017). The model dimension is set to 64. The number of heads in the Attention Mechanism is set to 1. The number of transformer blocks is set to 2. Our model adopts a learned (instead of fixed) positional embedding. The model size is similar to that in Ramaswamy et al. (2020), which is suitable for deployment on the user devices.

#### A.9.3 HYPERPARAMETERS

The number of epochs is set to 100. The batch size is chosen from $\{256, 512, 1024, 2048, 4096\}$. The learning rate is chosen from $\{10^{-3}, 3 \times 10^{-3}, 5 \times 10^{-3}, 7 \times 10^{-3}, 9 \times 10^{-3}\}$. The dropout rate is 0.2 for MovieLens and 0.5 for Amazon (due to its high sparsity). We use the Adam optimizer with a weight decay of $10^{-5}$.

### A.9.4 EVALUATION

HIT@$k$ measures whether the relevant (i.e., ground truth) item is present within the top-$k$ items in prediction list. It is a binary metric indicating the presence (hit) or absence of relevant items.

$$\text{HIT@}k = \begin{cases} 1, & \text{if the next item is in top-}k\text{ prediction} \\ 0, & \text{otherwise} \end{cases} \tag{32}$$

NDCG@$k$ measures the performance of a recommendation system based on the graded relevance of the recommended items. It is normalized based on the ideal order of items.

$$\text{NDCG@}k = \frac{\text{DCG@}k}{\text{IDCG@}k} \tag{33}$$

where

$$\text{DCG@}k = \sum_{i=1}^{k} \frac{2^{rel_i} - 1}{\log_2(i+1)} \tag{34}$$

Here, $rel_i$ is the relevance score of the item at position $i$ in the recommendation list. Namely, if the $i-th$ item (sorted by prediction score of the model) is equal to the ground truth, $rel_i = 1$ otherwise 0. IDCG@$k$ (Ideal Discounted Cumulative Gain at $k$) is the maximum possible DCG@$k$, obtained by placing the most relevant items in the top positions (i.e., the item with highest prediction score is equal to the ground truth). It's calculated similarly to DCG@$k$ but for the ideal order.

We adhere to the evaluation method advocated in Krichene & Rendle (2020), i.e., ranking all the items rather than adopting the sampled metrics where only a smaller set of random items and the relevant items are ranked. Note that under this evaluation method the accuracy value is greatly lower than the that obtained by sampled metrics, but is more consistent and meaningful.

## A.10    EXPERIMENTS ON ATTENTION SCORE DISTRACTION

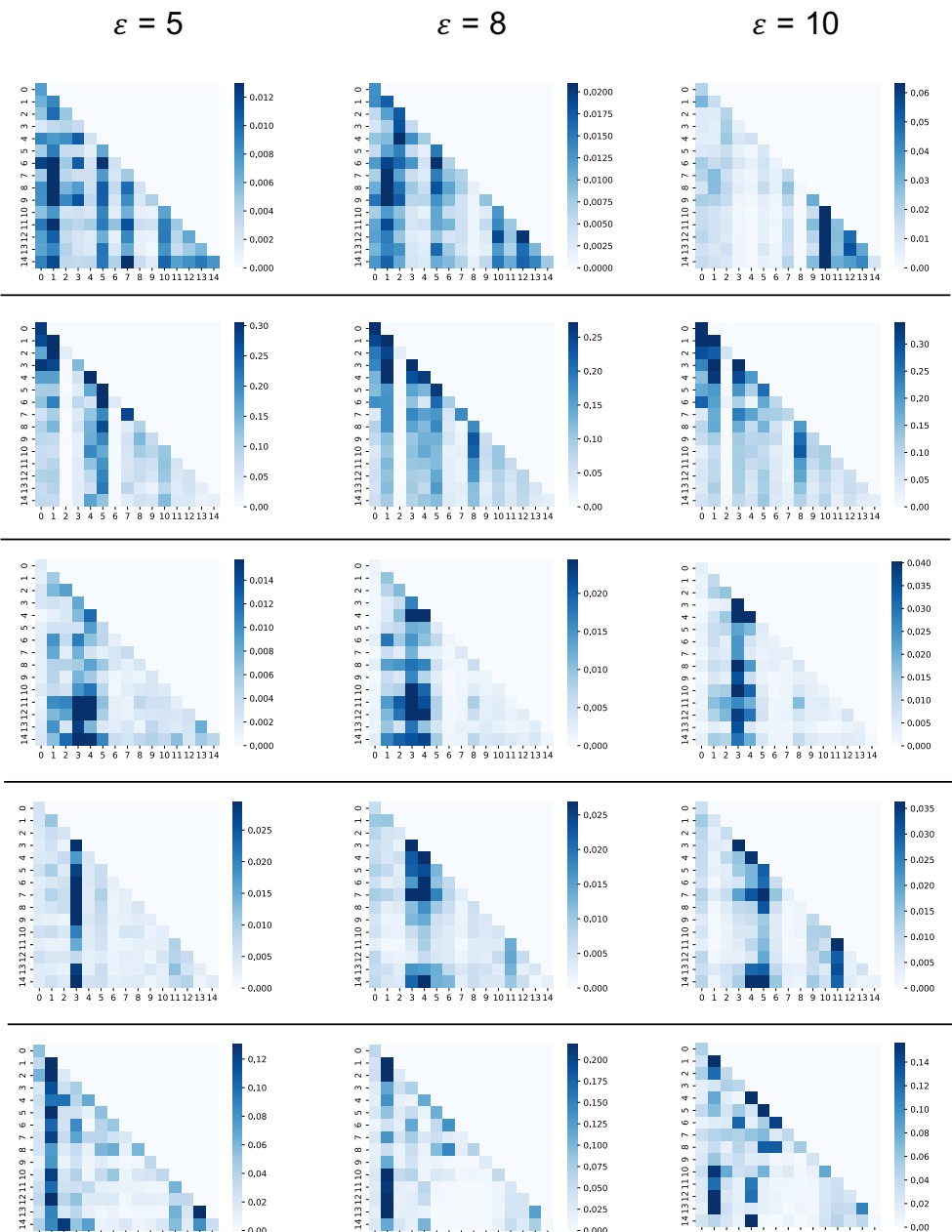

Figure 9: Visualization of attention score shift with varying $\varepsilon$ on MovieLens

To empirically demonstrate the attention score distraction phenomenon under private training, we randomly draw five sentences from MovieLens and visualize their attention matrices with $\varepsilon = 5, 8, 10$ (Figure 9). Recall that as the value of $\varepsilon$ increases (indicating a lesser amount of noise), the accuracy of the attention score improves (and consequently, the model performance) will be. With a larger $\varepsilon$ (10), the attention score distribution more closely aligns with the ground truth. It is clear from Figure 9 that due to the attention distraction caused by DP noise, when $\varepsilon$ is low, some tokens receive a larger amount of attention than they should have (compared with $\varepsilon = 10$), thereby resulting in suboptimal model performance.

