# OpenReview forum: "DPFormer: Learning Differentially Private Transformer on Long-Tailed Data"
_ICLR.cc/2024/Conference — Submitted to ICLR 2024_

### Official Review · Reviewer_eHtL · 2023-10-30

**Soundness:** 2 fair
**Presentation:** 2 fair
**Contribution:** 3 good
**Rating:** 6
**Confidence:** 4

**Summary:**

The paper introduces two main new modifications to differentially private training of transformers. They refer to it as DPFormer. The authors argue that two problems with differentially private training of transformers is i) the additional computational overhead of training with per-sample gradient (initially addressed by Li et. al. 2022 with Ghost clipping) and ii) a phenomenon the authors refer to as attention distraction. To resolve this, the authors propose Phantom clipping which also allows weight sharing (as opposed to Ghost Clipping)  and re-attention. Experiments on two datasets show improvement over some standard baseline algorithms like vanilla transformer, GRU, LSTM.

**Strengths:**

* __[S1]__ I appreciate the message that pre-training might not be possible for all tasks and it is important to develop algorithms that can work in settings without pre-training. This message has also been highlighted in [A,B].

* __[S2]__ It is important to show that modern DP algorithms can be run computationally efficiently, something which is often missed in works that show state-of-the-art accuracy at a huge computational cost. This problem is particularly pronounced for DP model-training where large batch sizes have somehow been used very commonly. To that effect, I appreciate the focus towards decreasing compitational cost.












[A]  "Considerations for differentially private learning with large-scale public pre-training." TPDP 2023.
[B]  "PILLAR: How to make semi-private learning more effective" TPDP 2023

**Weaknesses:**

* __[W1]__ If data is long-tailed, the hand-wavy definition of this means that some parts of the data are over-represented and some parts are under-represented. As such, if the objective is to get high average accuracy (or its variant for recommendation problem), then ignoring the low frequency elements shouldn't be a problem. This paper seems to be looking at average metrics but it mentions long-tailed data in the title.

* __[W2]__ The paper proposes a phenomenon called `attention distraction'. The suggestion is that tokens with high variance will be unfairly selected by the attention mechanism. But it is not very clear to me what this means formally. In addition, the theory seems to assume that every token's attention value is an independent gaussian, which is also unrealistic as all tokens are dependent on each other due to previous layers and I don't see why adding gaussian noise to gradients results in gaussian distribution for token activations. The whole section 4 assumes these two things but there isn't any empirical or theoretical justification for this.

* __[W3]__ The experimental comparison is not thorough at all. Please see question 3 below for a more detailed explanation of why I think that is the case.



Minor Comments
1. What do the metrics used here actually mean ? Maybe I missed it but I don't see a definition of ndcg@10 and HiT@10.
2. Figure 1 is not very understandable as there are terms like $a_s$ and $e_s$ which is not clarified.
3. In general, the text is very heavy which can be distracting to readers. I would recommend (personally) to have shorter sentences to the point.
4. Eq 1 isn't clear what G is.
5.

**Questions:**

1. Regarding long-tailed data [W1] cam the authors explain
    * motivation for why that is the central part of the story and where the long-tailed data comes into play.
    * any evidence that the proposed method specifically helps for long-tailed methods.
Perhaps the authors can draw some relation from [1] ?

2. Regarding [W2], can the authors provide evidence that ``attention distraction''
    * truly happens without re-attention mechanism in transformer. I understand Figure 9 is supposed to show this but I don't exactly understand what the two axis here is. I am also not comfortable drawing summary statistics about this as a wider phenomenon that what is shown in the five plots. To me it appears that this is just learning a biased classifier (i.e. not learning the true signal) that puts all its signal on one token (if I understand it correctly).
    * follows the same mathematical equations that Section 4 predicts it should.
    * Re-attention actually solves it ?

3. Both the datasets and baseline algorithms are fairly restrictive. Is there any reason apart from not having pre-trained models specialised for recommendation engines, why the experiments are limited to recommendation settings.
    * The methods should also provide benefits for NLP tasks (with and without pre-training).
    * Also, is there a comparison with Li et. al. or any other method for training transformers or recurrent models ? Such comparisons would be needed.
    * Can the authors also try using a general pre-trained transformer for this recommendation tasks ?

[1] "Does learning require memorization? a short tale about a long tail." Proceedings of the 52nd Annual ACM SIGACT Symposium on Theory of Computing. 2020.
[2] "How unfair is private learning?." In Uncertainty in Artificial Intelligence, pp. 1738-1748. PMLR, 2022.

---

> ### Author Response · Authors · 2023-11-20
>
> > Regarding long-tailed data [W1].
>
> **Reply:** Thanks for pointing out this perspective. We would like to make it clear that we made no attempt to deal with the fairness issue. Our motivation is long-tailedness in private learning will obstruct the training process (i.e., the attention distraction) and lead to (average) performance drop (i.e., low accuracy on both over-represented and under-represented data). Therefore, our goal in the task of 'improving private learning on long-tailed data' is to mitigate this additional issue and the ideal result is to recover the case of 'non-private (vanilla) learning on long-tailed data' (i.e., potentially low accuracy on under-represented data, but with a high average performance). We make no attempt to improve the model performance on under-represented data. Such goal can be achieved by combining existing method with our method.
>
>
> >  [W2] In addition, the theory seems to assume that every token's attention value is an independent gaussian, which is also unrealistic as all tokens are dependent on each other due to previous layers and I don't see why adding gaussian noise to gradients results in gaussian distribution for token activations. The whole section 4 assumes these two things but there isn't any empirical or theoretical justification for this.
>
> **Reply:** Thanks for pointing out. We would like to stress that these assumptions are commonly used and well-understood in the literature of variational inference and more broadly, bayesian deep learning. We directly adopt this methodolody from that area. The unfortunate fact is, (at this stage of history) we can only assume them being independent Gaussian and approximate it via moment matching. The development of more advanced techniques to help track the evolution of distribution through neural layers is beyond the scope of this work. In addition, we add a more detailed version of analysis of the attention distraction in Appendix A.3 in the revised paper.
>
> > why the experiments are limited to recommendation settings.
>
> **Reply:** We believe that Recommendation task is the most prevailing sequential task in the real-world that features the property of long-tailedness and lack of pre-trained model. Therefore, we target recommendation settings in the work.
>
> > Is there a comparison with Li et. al?
>
> **Reply:** We compare the efficiency of our phantom clipping with ghost clipping in Li et.al in Figure 3. For the accuracy aspect, the baseline transformer can be viewed as an instantiation of Li et.al since Li et.al did not propose algorithmically different methods.
>
> > Can the authors also try using a general pre-trained transformer for this recommendation tasks?
>
> **Reply:** To our knowledge, how to use a general pre-trained transformer for recommendation task is still a largely open problem. One cannot naively adopt a pre-trained transformer model because the input space is different (in recommendation task, token $i$ is associatedd with the item with ID $i$ while in NLP token $i$ is associated with some (sub)word defined by the tokenizer).

---

> > ### Comment · Reviewer_eHtL · 2023-11-22
> >
> > I thank the authors for their reply.
> >
> > My initial concerns (W1, W2, and W3) still remain though now I understand the authors' position on this, which is
> >
> > W1: the authors look at the setting of long-tailed data as opposed to improving the performance on the long tail. However, it is not yet clear to me what the the specific impact of the long tail is. I appreciate for the attempt to clarify but I think there is more work needed to clarify this.
> >
> > W2: Regarding the assumption, the authors admit that the assumption is strong and it is difficult to improve upon it. I understand the position and maintain that the assumption is too strong in my opinion for this particular setting. It may be more relevant in probabilistic modelling tasks.
> >
> > W3: The experimental baseline is still very narrow. The authors argue that it is not clear how to extend it but then I think this is something the paper should work towards: showing how widely applicable the method is as oposed to mentioning that this may not be widely applicable.
> >
> > I maintain my rating of borderline accept.

---

### Official Review · Reviewer_v4h1 · 2023-10-31

**Soundness:** 2 fair
**Presentation:** 3 good
**Contribution:** 3 good
**Rating:** 5
**Confidence:** 3

**Summary:**

This paper studies the differential privacy in transformers. They identify two key challenges as heavy computation due to per-sample gradient clipping, and unintentional attention distraction within the attention mechanism. They propose DP- Former, equipped with Phantom Clipping and Re-Attention Mechanism, to address these challenges. The theoretical analysis shows that DPFormer can reduce computational costs during gradient clipping and effectively mitigate attention distraction. Empirical results on two real-world recommendation datasets with varying degrees of long-tailedness, showing its significant improvement in terms of efficiency and effectiveness.

**Strengths:**

This paper give DPformer to address the challenges of heavy computations and attention distraction.
Originality: The proposed methods addressed the two difficulties, it’s novel and has good performance. Phantom Clipping inherit the basic idea of Ghost Clipping, obtain the per-sample gradient norm without the need for instantiating the per-sample gradient. p
Quality: This methods performs well in experiments. But I doubt that whether the re-attention method after DP process can leak privacy?
Clarity: The logic structure of this paper is clear
Significance: These two problems are very important problems for differential privacy in Transforms, so this kind of work is necessary

**Weaknesses:**

The correctness of this method.
This paper uses Phantom clipping and Re-attention mechanism to overcome the difficulties of heave computation and attention distraction. The most important thing is to make the whole mechanism be differentially private and have high utility. However, they authors didn't analyze the privacy of this mechanism, especially after the re-attention. It seems this process will leak privacy.

**Questions:**

What is the privacy guarantee for this method?
What do NDCG@10 and HIT@10 mean?

---

> ### Author Response · Authors · 2023-11-20
>
> > What is the privacy guarantee for this method?
>
> **Reply:** Thanks for pointing out. We have explicitly made the privacy statement in the revised version (Claim A.5). In particular, the privacy guarantee is inherited from that of DP-SGD. Details follow.
>
> Phantom clipping is an efficient implementation of gradient clipping, having the same functionality as vanilla per-sample clipping (i.e., the same input-output behavior). Therefore, we can think of it as the vanilla gradient clipping when we analyze the privacy.
>
> For Re-Attention mechanism, observe that the whole computation process only depends on the single data sample (i.e., we do not create dependency among samples between a mini-batch, thus violating DP like Batch Normalization). Another way to understand this is we can do whatever we like in the forward pass as long as the computation associated with one input sample does not rely on other input samples of the mini-batch. (A caveat is discussed in Remark A.2 of Appendix.A.4) Therefore, the private analysis reduces to DP-SGD: By clipping the gradient we can successfully bound the per-sample sensitivity (and add the calibrated noise to guarantee DP).
>
>
> > What do NDCG@10 and HIT@10 mean?
>
> **Reply:** Thanks for pointing out. They are metrics commonly used in recommendation systems to measure the prediction accuracy. They can be simply understood as Top-10 accruacy and weighted Top-10 accuracy as in the classification task. We have included the definitions in Appendix A.9.4 in the revised paper.

---

> > ### Comment · Reviewer_v4h1 · 2023-11-23
> > **Thanks for the response**
> >
> > Authors response resolved my doubts about the privacy. I keep my score because it's hard for me to follow and check the technological and mathematical principles. I agree that this paper works on an important research question and gets good results in their experiments, but limited to recommendation dataset.
> > ​

---

### Official Review · Reviewer_SzC2 · 2023-10-31

**Soundness:** 3 good
**Presentation:** 3 good
**Contribution:** 3 good
**Rating:** 6
**Confidence:** 4

**Summary:**

This paper introduces two treatments for training transformer models with DP-SGD on long-tailed data. The first is phantom clipping which extends ghost clipping by handling parameter sharing between input and output layer (a common practice). Phantom clipping effectively enabled more efficient DP training of transformer models in practice. The second is a re-attention mechanism which rescale the attention scores in the transformer models by their error variances due to DP noise. The methods are evaluated on two benchmark recommendation datasets and showed better performance compared to the DP-SGD baseline.

**Strengths:**

- This paper is well-motivated as transformer models are used in many ML applications and being able to train it with strong privacy guarantees is important for these models to scale further.
- The methods are driven with the empirical observations of the shortcomings of DP-SGD on transformer models. Parameter sharing of embedding parameters is such a common thing to do but was not considered in prior work on speeding up DP-SGD. Also DP is known for generalizing worse on the tail and the re-attention mechanism is a simple yet effective workaround.

**Weaknesses:**

- The experiments, as the authors acknowledged, were conducted on the smaller scale models. Also only recommendation tasks were considered. It is unknown whether these methods will be effective on larger image or text models.
- Some baselines for training with DP are missing where these methods also improve the performance of models for long-tailed data: [1] proposed a way to add DP noise adaptively according to the geometry of the loss space, which can effectively enable smaller noise on long-tailed data. [2] similarly uses auxiliary information such as the frequency of the vocabulary to apply DP noise.
- Describing section 4 as an algorithm would be easier to read.

References

[1] Asi, Hilal, et al. "Private adaptive gradient methods for convex optimization." International Conference on Machine Learning. PMLR, 2021.

[2] Li, Tian, et al. "Private adaptive optimization with side information." International Conference on Machine Learning. PMLR, 2022.

**Questions:**

- Is Figure 3 based on Ghost Clipping or Book-Keeping [1]? BK is supposed to be more efficient than Ghost Clipping.
- In Equation 7, are the subscripts on the right-hand side supposed to be $X^{(l-1)}$?

References

[1] Bu, Zhiqi, et al. "Differentially private optimization on large model at small cost." International Conference on Machine Learning. PMLR, 2023.

---

> ### Author Response · Authors · 2023-11-20
>
> > Is Figure 3 based on Ghost Clipping or Book-Keeping [1]? BK is supposed to be more efficient than Ghost Clipping.
>
> **Reply:** Yes, exactly. We use implementation in BK for the baseline.
>
> > In Equation 7, are the subscripts on the right-hand side supposed to be $X^{(l-1)}$.
>
> **Reply:** Yes. Sorry for the confusion.
>
> > Some baselines for training with DP are missing where these methods also improve the performance of models for long-tailed data
>
> **Reply:** Thanks for pointing these great works. We discuss them in the related work in Appendix A.1 in  the revised paper. From our understanding, these works are operating on the optimizer level, while our method does not assume any specific optimizer and uses the underlying optimizer in a black-box manner. Therefore our method is orthogonal to them and can be potentially combined together.
>
> > Describing section 4 as an algorithm would be easier to read.
>
> **Reply:** Thanks for your suggestion. We add an algorithm description in the Appendix A.7.

---

### Official Review · Reviewer_FhGH · 2023-11-03

**Soundness:** 2 fair
**Presentation:** 2 fair
**Contribution:** 2 fair
**Rating:** 3
**Confidence:** 3

**Summary:**

This paper identifies two key challenges in learning differentially private Transformers, i.e.,
heavy computation overhead due to per-sample gradient clipping and attention distraction due to
long-tailed data distributions. The authors then proposed DPFormer, equipped with Phantom Clipping and
Re-Attention Mechanism, to address these challenges. Empirical results are also provided to justify the new design.

**Strengths:**

The paper aims at solving issues and improving the performance of DP transformers. Personally, I think this problem is very important and a key step towards enabling large model training under differential privacy. Given the popularity of large language models and transformers, this problem should be interesting to both DP community and LLM community.

**Weaknesses:**

Although the problem at which the paper aims is very important, the paper does not provide a satisfactory answer and leave many questions unanswered. Most importantly, this paper needs to greatly improve its clarity: there are many places with vague languages or unstrict math, making it very hard for the reader to follow or evaluate the paper's quality. Personally speaking, there is still a long way to go before the paper can be accepted.

Here are some specific suggestions/confusion:

Phantom clipping:
1. All the notations in Claim 3.1 are used without being carefully defined. I do not find it enough to just define them by Figure 1 or simple languages in Claim 3.1. Could we have some strict math like section 2.2 in https://arxiv.org/pdf/2205.10683.pdf?
2. I am not convinced why the phantom clipping has such great advantages over ghost clipping. Observing Claim 3.1, it has exactly the same idea (calculating the norm and avoiding gradient instantiation) with ghost clipping. Specifically, I do not understand why the BM^2 complexity is inherent to the ghost clipping under a careful implementation. With that in mind, I am concerned the paper may over-claim the advantages.
3. Looking at the results from other papers  (e.g., https://arxiv.org/pdf/2205.10683.pdf) applying ghost clipping to transformers, the ghost clipping has shown a much better memory effIciency. I do not quite understand the gap between Figure 3 and those results.

Re-attention
Generally speaking, the idea of re-attention is interesting. However, there are many vague places in both motivation and estimation, making me hard to feel convinced. Specifically,
1. The math in the motivation is not very strict. Please make some revise. Furthermore, there are many other sources of randomness in model training except for DP noise. Please specify them in the conditional distribution.
2. In section 4.1, the paper mentions that the attention distraction mainly happens when different parameters have different variance. However, in DP-SGD, the noise with the same std is added to each parameter. Could the authors explain about this?
3. Section 4.2.2 is very hard to follow. For me, it just reads like a mixture of several techniques without a clear explanation. I have several confusions. For example, is the assumption realistic assuming i.i.d. Gaussian for each dimension in parameter? Furthermore, I do not quite follow how to estimate the mean and std of the start point.

Experiments
1. Since the paper is targeting large models, it is better to have some setting for fine-tuning instead of training from scratch. Also, revealing some results when eps = infty will also be helpful.

**Questions:**

Please refer to weakness.

---

> ### Author Response · Authors · 2023-11-20
>
> > The math is not very strict. Please make some revise.
>
> **Reply:** Thanks for your suggestion. In the revised paper, we give the detailed version of phantam clipping in Appendix A.2 and motivation behind re-attention in Appendix A.3.
>
> > I am not convinced why the phantom clipping has such great advantages over ghost clipping. I do not understand why the BM^2 complexity is inherent to the ghost clipping under a careful implementation.
>
> **Reply:** $BM^2$ complexity is provided in the ghost clipping paper. Our contribution is to support parameter sharing along with a 'careful implementation'.
>
> > The ghost clipping has shown a much better memory effIciency. I do not quite understand the gap between Figure 3 and those results.
>
> **Reply:** Our results are not contradictory to existing results. Previous results are obtained when using a large model while we use a relatively small model.
> Note that the advantage of ghost clipping vanishes as model size becomes smaller.
>
> > The attention distraction mainly happens when different parameters have different variance. However, in DP-SGD, the noise with the same std is added to each parameter.
>
> **Reply:** A key observation is, for embedding-based models, not all embedding parameters are equal. As a thought experiment, let the embedding layer be parameterized by $E$ (of shape $m \times d$). Suppose we add a dummy embedding $e_{m+1}$ into $E$ to obtain $E'$ (of shape $(m + 1) \times d$) and start DP training, but no data is associated to that dummy token m+1. During each iteration t, we add Gassian noise $n_t$ to the whole $E'$. After $T$ iterations, it is obvious that e_m+1 will become very large (i.e., $e_{m+1} = n_1 + n_2 + ....+ n_T$, distributed as Gaussian with large variance $T \sigma$). Our definition of effetive error (section 4.2.1) is intended to capture this.
>
>
> > Is the assumption realistic assuming i.i.d. Gaussian for each dimension in parameter?
>
> **Reply:** Note that Gaussian for each dimension in model parameter is not an assumption. It is induced by the DP Gaussian noise (fixing all other randomness). We presume the reviewer's concern is regarding 'use an approximating distribution $q$ to approximate the  distribution $p$, where $q$ follows a indepent Gaussian distribution'. This is a commonly used and well understood assumption in the literature of variational inference and more broadly, Beyesian deep learning, since $p$ is computationally intractable. We can only approximate it using a Gaussian via moment matching. The development of more advanced techniques to help track the evolution of distribution through neural layers is beyond the scope of this work.
>
>
> > I do not quite follow how to estimate the mean and std of the start point?
>
> **Reply:** For example, if the training sequence is [1, 2, 3], then the starting point is $[E_1, E_2, E_3]$ (where $E_i$ is the embedding of token i). The std is estimated by error instantiation (see section 4.2.1). The mean is estimated by itself, $E_i$ (note that access to this noisy parameter can be
> interpreted as a single sampling opportunity from its underlying Gaussian distribution, which can then be viewed as a one-time Markov Chain sampling. Therefore, the noisy parameter can serve as an estimate of its mean).
>
> > Since the paper is targeting large models, it is better to have some setting for fine-tuning instead of training from scratch. Also, revealing some results when eps = 0 will also be helpful.
>
> **Reply:** We actually target small model in this paper, which is justified by considering the computing resource of end users (preferable for privacy protection through local inference.) Note that for recommendation tasks, how to use a general pre-trained transformer for recommendation task is still a largely open problem. One cannot naively adopt a pre-trained transformer model because the input domain is different (in recommendation task, token $i$ is associatedd with the item with ID $i$ while in NLP token $i$ is associated with some (sub)word defined by the tokenizer). Results when eps = 0 is nearly 0 for both metrics.

---

### Meta-Review · Area_Chair_wwAw · 2023-12-05

**Metareview:**

The paper proposes two tricks, phantom clipping and re-attention mechanism, to make learning small transformers from scratch under DP more efficient both computationally as well as statistically.

Strengths: improving the efficiency of training transformers is very relevant and timely. The experimental results appear promising.

Weaknesses: the paper does a poor job at communicating the limitations of the method, many of which have only emerged in response to reviewer comments and are not reflected in the paper. Major assumptions such as knowing the token marginal distribution are hidden in the Appendix (Remark A.2 in Section A.4).

The results claim massive speedup for Phantom Clipping relative to Ghost Clipping, although apparently this only happens in specific circumstances. The results on phantom clipping would clearly need to be extended to cover a broader range of transformer applications (including large models and fine tuning) to allow the readers to easily evaluate the scope of the contribution.

The presentation of Re-Attention Mechanism would also need to be expanded to clearly highlight its limitations and evaluate when it is useful. Assumptions such as knowing the token marginal distribution need to be made clear. Given that the paper claims to target long-tailed data, I have a feeling that this distribution will not be so easy to evaluate accurately. The authors should test what happens when the distribution is not perfectly known.

I wonder if the chosen hyperparameter ranges favor DPFormer? It seems that the optimal hyperparameters for DPFormer are in the middle of the range while the optimal hyperparameters for baselines are at the extreme. This suggests that the baseline might perform even better if the range were extended. This should be clarified.

Results should be reported for levels of DP that provide actual formal protection, i.e. $\epsilon=1$.

**Justification For Why Not Higher Score:**

The average reviewer opinion favors rejection. Based on my own evaluation, it feels that the paper is overselling the proposed method.

**Justification For Why Not Lower Score:**

N/A

---

### Decision · Program_Chairs · 2024-01-16

Reject